# Emotion Regulation Strategies and Psychological Well-Being in Emerging Adulthood: Mediating Role of Optimism and Self-Esteem in a University Student Sample

**DOI:** 10.3390/bs15070929

**Published:** 2025-07-09

**Authors:** Hugo Sanchez-Sanchez, Konstanze Schoeps, Inmaculada Montoya-Castilla

**Affiliations:** Department of Personality, Assessment and Psychological Treatment, University of Valencia, Av. Blasco Ibáñez 21, 46010 Valencia, Spain; hugo.a.sanchez@uv.es (H.S.-S.); inmaculada.montoya@uv.es (I.M.-C.)

**Keywords:** psychological well-being, emotion regulation strategies, self-esteem, optimism, emerging adulthood, emotion development, mediation model

## Abstract

Emerging adulthood is a critical phase for emotional development and mental health. Psychological well-being has been associated with using emotion regulation strategies as well as high self-esteem and high optimism. The objective of this study was to examine the mediating role of self-esteem and optimism in the relationship between emotion regulation and psychological well-being in the context of the challenges associated with emerging adulthood. The study was conducted with the participation of 771 university students (M = 20.38, SD = 2.01, 73.3% female), who completed questionnaires, including the Psychological Well-Being Scales (PWBS), the Cognitive Emotion Regulation Questionnaire (CERQ-18), Rosenberg’s Self-Esteem Scale and the Optimism Questionnaire (COP). The results indicate a positive correlation between the adaptive strategies of emotion regulation and the dimensions of psychological well-being, as well as a positive correlation with self-esteem and optimism. A mediation model was tested with an adequate model fit, examining both direct and indirect effects. The model identifies planning, positive reappraisal, and catastrophizing as the most significant emotion regulation strategies, while also emphasizing the importance of some dimensions of well-being, such as self-acceptance, mastery of the environment, and life purpose. Furthermore, the findings illustrate the role of self-esteem and optimism as mediators in these relationships. The study concludes with an analysis of the theoretical and practical implications of the findings in the context of the difficulties associated with emerging adulthood where individuals define their identity, goals and purposes in life and their personality becomes more consistent.

## 1. Introduction

### 1.1. Emerging Adulthood: A New Developmental Stage

Over the past two decades, there has been a notable increase in the literature focusing on emerging adulthood from a human development perspective. Emerging adulthood, defined as the period between ages 18 and 29, is marked by a range of new developmental challenges that can significantly impact emotion and psychological well-being ([9]; [21]). This phase is often characterized by prolonged educational and training trajectories, largely shaped by the demands of contemporary economic and industrial systems ([5]). As a result, the traditional, normative experiences associated with this age range have shifted substantially for newer generations ([6]). The accumulation of personal, academic, financial, and social pressures during this time can negatively affect mental health, making it a period of heightened vulnerability and increased prevalence of disruptive behaviors ([42]).

According to [2] ([2]), emerging adulthood is experienced as a distinct developmental stage and defined by identity exploration, experimentation, instability, self-focus, other-focus, and a feeling in-between adolescence and adulthood. This identity exploration can provoke confusion and fear, particularly among individuals with low self-esteem, while instability may be exacerbated by the unpredictable nature of the transitions faced. Moreover, the psychological tension stemming from not fully identifying as either an adolescent or an adult has been associated with elevated levels of anxiety and depression, which may in turn hinder autonomy and exploratory behaviors ([6]; [12]). During this complex period, high levels of stress and uncertainty are common and have been linked to an increased risk of developing psychiatric disorders ([7]; [12]). Understanding how young adults navigate these challenges is crucial, as their experiences during this time can shape their long-term mental health and development ([9]).

### 1.2. Psychological Well-Being in Emerging Adulthood

The period of emerging adulthood is characterized by substantial psychosocial changes that can have a considerable impact on both mental health and overall well-being ([12]). Although numerous studies have identified elevated stress levels and emotion difficulties as common experiences during this period ([4]), it is crucial to recognize that this phase can also be a time for individuals to engage in processes that promote psychological well-being ([62]). Therefore, emerging adulthood is characterized by a duality of vulnerability to mental health challenges and the capacity for positive psychological development ([34]).

Well-being has been identified as particularly relevant during emerging adulthood ([64]). Well-being is a central psychological concept, explored from two main theoretical perspectives. The eudaimonic perspective focuses on personal growth, developing virtuous traits, and pursuing meaningful life goals ([55]; [56]). In contrast, the hedonic perspective emphasizes the pursuit of pleasure and the avoidance of pain ([16]; [17], [18]). The present research focuses on the eudaimonic perspective, more specifically on Ryff’s model of psychological well-being ([54], [55]), which is a widely developed integrative model based on humanistic, existential, and positive psychology ([35]).

Ryff’s model proposes that true well-being is not merely the absence of psychological problems or the pursuit of happiness, but rather a state of living in harmony with oneself, with a focus on leading a purposeful and meaningful life ([16]). Recent research in positive psychology has sought to identify variables that contribute to well-being across different stages of development ([47]; [61]). It has been consistently shown that well-being in emerging adulthood is associated with healthy behaviors that enhance mental health and increase life expectancy ([64]).

### 1.3. Emotion Regulation Strategies and Mental Health

A fundamental aspect of well-being during emerging adulthood is the capacity to regulate emotions effectively. Emotion regulation strategies, a key predictor of well-being in both clinical and research settings ([10]), refers to a diverse set of processes that help individuals manage their emotion responses to achieve specific goals and adapt to their environment ([32]). The process model of emotion regulation ([30], [31]) suggests that people use both antecedent-focused strategies, such as selecting or modifying situations and cognitive reappraisal, and response-focused strategies like emotional suppression. These strategies play a critical role in maintaining psychological balance and well-being and they can be categorized as adaptive or maladaptive. Adaptive strategies, such as cognitive reappraisal and planning, are associated with positive outcomes, including personal growth, environmental mastery, and a sense of life purpose ([10]; [28]). For example, cognitive reappraisal, which means that individuals reinterpret a stressful situation in a more positive light, is one of the strongest predictors of well-being ([33]; [43]; [63]; [67]). Maladaptive strategies like emotional suppression or catastrophizing can have detrimental effects, leading to lower psychological well-being and a more negative evaluation of life ([1]; [25]). Catastrophizing may lead people to overestimate the probability, risk, and severity of negative events, and it may reduce satisfaction with preventive behaviors in the face of negative events. This facilitates the perpetuation of these dysfunctional beliefs and contributes to the maintenance of anxiety disorders ([26]). Thus, the way individuals regulate their emotions has a profound impact on their overall psychological health ([25]).

### 1.4. Personality Traits: Self-Esteem and Optimism

The available research evidence indicates that the most significant transformations and changes occur during emerging adulthood, as do the processes or mechanisms that influence personality traits and individuals’ behavioral, cognitive and emotion development ([49]). This research has led to the development of integrative models such as the Neo-Socioanalytic Model of Personality that provides a framework for understanding the relationship between personality traits, human development and the study of emerging adulthood ([49]; [50]). In particular, the present paper highlights self-esteem and optimism as key contributors to psychological well-being and emotion regulation in emerging adulthood.

Research has shown that difficulties in emotion regulation are often linked to low self-esteem and pessimism ([8]; [68]). Self-esteem, defined as an individual’s overall positive or negative attitude towards themselves, plays a crucial role in how they perceive and respond to life’s challenges ([52]). Individuals with high self-esteem tend to view themselves as valuable and worthy, fostering resilience and positive mental health. Conversely, those with low self-esteem may struggle with self-rejection and dissatisfaction, which can complicate effective emotion regulation and well-being. Optimism, another critical factor, refers to a general tendency to expect positive outcomes in the future ([13], [14]). Optimistic individuals are more likely to use adaptive strategies like cognitive reappraisal and acceptance, which contribute to higher levels of well-being ([68]). On the other hand, those with a more pessimistic outlook are prone to employing maladaptive strategies, such as avoidance or suppression, which can negatively affect their mental health ([1]; [45]). Research consistently supports the role of optimism as a predictor of psychological well-being across diverse contexts and populations ([11]; [23]; [57]). Emerging adulthood is characterized by a sense of possibility and optimism and thus it provides a unique context to explore these dynamics further. Despite the uncertainties and challenges of this developmental stage, many emerging adults maintain a hopeful outlook on their future, which can act as a protective factor in their pursuit of personal goals and life satisfaction ([48]). This optimism fosters resilience and supports continuous personal development, even in the face of setbacks and difficulties.

### 1.5. Objective and Hypotheses of the Study

This study is essential for advancing our understanding of the mechanisms linking emotion regulation and psychological well-being during emerging adulthood, a developmental stage marked by significant personal and emotional challenges in contemporary society. Investigating how self-esteem and optimism relate to emotion regulation and well-being provides valuable insights that can inform efforts to enhance mental health outcomes in this population. Furthermore, identifying the most influential variables in promoting well-being is particularly relevant given the complex and multifaceted demands faced by emerging adults today.

The aim of this study was to explore how self-esteem and optimism mediate the relationship between emotion regulation strategies and psychological well-being in emerging adults. We hypothesize that individuals with higher levels of self-esteem and a positive optimism about the future are more likely to engage in adaptive emotion regulation strategies, which, in turn, enhances their well-being. To test this, we formulated four hypotheses: H1: Emotion regulation strategies (X) will be associated with psychological well-being (Y); more specifically, adaptive strategies will be positively related, and maladaptive strategies will be negatively related to well-being dimensions. H2: Emotion regulation (X) will be positively related to self-esteem (M1) and optimism (M2); more specifically, adaptive strategies will be positively related, and maladaptive strategies will be negatively related to self-esteem and optimism. H3: Self-esteem (M1) and optimism (M2) will be positively related to psychological well-being (Y). H4: Self-esteem (M1) and optimism (M2) will mediate the relationship between emotion regulation (X) and psychological well-being (Y) (see Figure 1).

## 2. Materials and Methods

### 2.1. Participants

The dataset consisted of 1074 records, which were refined according to pre-established criteria. First, based on the definition by ([2]), only participants aged between 18 and 29 years were included, leading to the exclusion of 21 subjects who did not meet this age criterion. Subsequently, an infrequency scale ([24]) was applied, resulting in the removal of 3 additional subjects. Furthermore, only responses that were at least 80% complete were considered, which led to the exclusion of 279 subjects. After applying these criteria, a total of 771 participants were included in the study (M = 20.38; SD = 2.01; 73.3% female).

Most of the participants (94%) were undergraduate students, with a smaller proportion (6%) of PhD students. The student population comprised individuals from various academic disciplines, including health sciences (72%), which encompasses disciplines such as psychology, medicine, and nutrition; social and legal sciences (11%); information technology (10%); and a small percentage from industrial sciences and aeronautics (7%). With regard to economic resources, 50.7% indicated that they are sufficient, followed by somewhat lower (9.8%) and somewhat higher (19.6%) figures. The data indicate that 75.7% of the participants were studying, 23.7% studied and worked, and 0.4% worked. In total, 61% of the participants reside with their family of origin, 32.4% in cohabitation, 2.9% with a partner, and 2.3% by themselves. Income is derived from familial sources for 58.2% of participants, familial and occupational sources for 25.6%, subventions for 8%, and employment for 7.8%.

The sampling was obtained by convivence from various universities in the region. The survey was distributed by university professors, both during classroom sessions and via Email. The completion duration was approximately 20 min. Participants took part in a raffle for gift cards as an incentive. To a priori determine an appropriate sample size, a calculation was carried out using G Power Software V.3.1.9.7 ([22]). The parameters were based on a linear multiple regression analysis, effect size f2 of 0.02, with a confidence level of 95%, a statistical power of 0.8, and 5 predictors. The calculated sample size was determined to be 647 individuals, which is less than the sample obtained.

### 2.2. Instruments

The instruments chosen for this research were selected because of their adaptation and validation for the population in which the study was conducted. In addition, the instruments have demonstrated high levels of validity and reliability in previous research.

To identify the cognitive strategies employed to regulate emotion responses, the short version of the Spanish Cognitive Emotion Regulation Questionnaire (CERQ-18) was be used ([28]; [36]). The test comprises nine strategy categories, self-blame, other-blame, rumination, catastrophizing, putting into perspective, positive refocusing, positive reappraisal, acceptance, and planning. The questionnaire comprises 18 items, each of which is answered on a 5-point Likert scale (1 = Never, 5 = Always). The reliability of the scale indices was demonstrated by the Spanish adaptation, with coefficients for adaptive strategies (α = 0.84) and maladaptive strategies (α = 0.72) ([36]). The indices observed in this study were comparable, with α = 0.78, ω = 0.65 in adaptive strategies, and α = 0.72, ω = 0.77 in maladaptive strategies.

The Rosenberg Self-Esteem Scale was utilized to assess self-esteem. An elevated score on the scale is indicative of a higher level of global self-esteem. The scale comprises 10 items, each of which is rated on a 4-point Likert scale (1 = Strongly disagree; 4 = Strongly agree). In the Spanish version of the scale, Cronbach’s α coefficient was α = 0.84. ([44]). This is like the value observed in our study: α = 0.9, and ω = 0.9.

The assessment of optimism was conducted using the Optimism Questionnaire ([46]). A higher score on this scale indicates a greater dispositional optimism. The items (9) are presented on a Likert scale, with responses ranging from 1 (indicating a strong disagreement) to 5 (indicating a strong agreement). The scale has demonstrated satisfactory psychometric properties, meeting the criteria for reliability (α = 0.84) and construct validity ([46]). Similarly, in the present study, we observed a value of α = 0.9, and ω = 0.9.

The Spanish version of the Psychological Well-Being Scales (PWBS) was employed to assess psychological well-being. A score that is higher on each scale indicates a higher level of psychological well-being in the following dimensions: self-acceptance, positive relations with others, autonomy, environmental mastery, purpose in life, and personal growth. The instrument comprises a total of 29 items, each of which is rated on a 6-point Likert scale (1 = Strongly agree, 6 = Strongly disagree). The psychometric properties of this instrument were found to be satisfactory in both its original version (α = 0.78 to α = 0.81) ([53]) and in its Spanish validation (α = 0.70 to 0.84) ([19]). In our analyses, indices ranging from ω = 0.63 (environmental mastery) to 0.86 (self-acceptance) were identified.

Finally, the Oviedo Infrequency Scale was used ([24]). This is a 12-item self-report instrument utilizing a 5-point Likert-type scale. Its objective is to identify participants who respond randomly, pseudo-randomly, or dishonestly. Participants with more than three incorrect responses on this test were excluded from the sample.

### 2.3. Procedure

The procedure was carried out by the ethical principles outlined in the 1964 Declaration of Helsinki ([38]). The study was approved by the Ethical Committee of the University where the research was conducted. Prior to data collection, all participants provided informed consent, and they were informed that they were free to withdraw from the study at any time, without giving any reason.

### 2.4. Statistical Analyses

The analyses were conducted using Jasp (version 0.18.3) and SPSS (Version 28.0.1.1 (14). The mediation analyses were conducted following the AGREMA guideline ([40]). The AGREMA is a checklist based on evidence and consensus. It has been developed to provide recommendations for studies reporting statistical mediation analyses ([15]). Preliminary analyses were performed, calculating descriptive statistics for the variables of interest, including mean deviation, measures of asymmetry, the internal consistency indicators Omega and Cronbach’s alpha.

To examine the relationships between emotion regulation, psychological well-being, optimism and self-esteem, we conducted bivariate Pearson correlation analyses. Although the Kolmogorov–Smirnov tests indicated significant deviations from normality for all variables, the large sample size (*N* = 771) supports the robustness of parametric analyses, in line with the Central Limit Theorem ([41]). Additionally, visual inspection of histograms and Q–Q plots suggested approximately symmetric distributions with no extreme skewness or kurtosis, further justifying the use of Pearson’s correlation coefficients.

Based on significant correlations with an effect size greater than 0.3, we selected the variables that showed a strong association with each other to construct the mediation model. Furthermore, variables based on the existing literature were considered to facilitate the inclusion of variables that demonstrate a significant association ([65]). To test hypothesis 4, we conducted mediation analyses, using the model fit indicators recommended by [37] ([37]), and by testing direct, indirect and total effects, bootstraps and confidence intervals were estimated.

During the preparation of this work, the authors used Deepl AI (V. 1.49.0) and ChatGPT (V GPT-3.5) to enhance the academic writing style and to improve the translation of academic texts.

## 3. Results

### 3.1. Descriptive Statistics and Correlations

The detailed descriptive statistics, encompassing means, standard deviations, and the distribution of scores for all variables analyzed, are thoroughly presented in Table 1. It was found that the relationships were of a negative direction with maladaptive strategies and positive direction with adaptive strategies. The effect size was found to be higher than 0.3 for planning, positive reappraisal, and catastrophizing in all dimensions of well-being, except for positive relations with others.

In terms of correlations between emotion regulation strategies and the mediating variables, a correlation between self-esteem and optimism greater than 0.3 was identified, and for r oscillates it was between 0.43 and 0.53. Similarly, the correlations between self-esteem and optimism and the dimensions of well-being were all positive, with medium–high correlation indices for r oscillates between 0.35 and 0.79. (Table 2). Self-esteem and optimism showed strong positive correlations between 0.63 and 0.79 with dimensions of well-being such as self-acceptance, environmental mastery, and purpose in life. In contrast, a significant negative correlation, greater than −0.4, was observed between maladaptive strategies, specifically catastrophizing, and the dimensions of self-acceptance and environmental mastery, as well as with self-esteem and optimism. Conversely, adaptive strategies, such as acceptance and positive reappraisal, demonstrated positive correlations nearing 0.5 with the dimensions of well-being self-acceptance and life purpose, as well as positive correlations around 0.4 with self-esteem and optimism.

### 3.2. Model Fit

The model fit indices were, in general, considered to be acceptable. The CFI was 0.87, a value slightly below the conventional threshold of 0.90, but close enough to indicate an acceptable level of fit ([37]). The RMSEA was 0.76, with values below 0.8 generally considered to indicate an acceptable fit ([37]). The SRMR was 0.10, indicating that the model exhibits a reasonable degree of fit with the data. The R^2^ values for the well-being dimensions were found to be appropriate, with values of 0.64 for self-acceptance, 0.45 for environmental mastery, and 0.48 for purpose in life. In terms of the mediating dependent variables, the R^2^ values for self-esteem and optimism were 0.31 and 0.43, respectively.

It is important to note that not all possible relationships among the variables were considered in the final analysis. Instead, the focus was placed on the three emotion regulation strategies that exhibited the strongest correlations, along with the three well-being dimensions with the highest values. This selection reduced the model from 17 variables to 8. While this simplification may enhance the robustness and interpretability of the model, it also carries potential implications for its structural integrity due to the exclusion of relevant relationships. Therefore, a careful evaluation of the model fit indices is essential to assess its adequacy and ensure that key associations have not been overlooked.

### 3.3. Direct Effects

The results of the mediation model demonstrated a significant direct effect of emotion regulation strategies on the dimensions of psychological well-being. All direct effects were statistically significant, except for positive reappraisal with environmental mastery and catastrophizing with self-acceptance, and a direct positive effect between catastrophizing and life purpose was identified (Table 3).

Positive reappraisal had a significant direct effect on purpose in life. Planning had a significant direct effect on self-acceptance and environmental mastery, as well as on purpose in life. Catastrophizing demonstrated a significant inverse index with environmental mastery and, contrary to hypothesis 2, a positive relationship with purpose in life (Table 3).

### 3.4. Indirect Effects

The indirect effects demonstrate that emotion regulation strategies have an impact on psychological well-being through the influence of optimism and self-esteem. This is evidenced by the fact that all indirect effects were statistically significant, as estimated by both the bootstrap confidence interval methods.

The association between positive reappraisal on self-acceptance is mediated by self-esteem and optimism. Furthermore, planning demonstrated indirect effects on environmental mastery through self-esteem and optimism. Similarly, catastrophizing had a negative impact on self-acceptance through self-esteem and optimism, and a negative effect on purpose in life through self-esteem and optimism (Table 3).

### 3.5. Total Effects

The results indicated that all effects were statistically significant. Of the total effects, the largest were observed for planning through self-acceptance, followed by planning through environmental mastery with the same index, and then again, planning through purpose in life (Table 3).

Complete mediation was found between catastrophizing and self-acceptance, similar to positive reappraisal and environmental mastery, with self-esteem as an optimism-like mediator. All other mediations were partial, and both the direct and indirect effects were significant. As evidenced by the results, both self-esteem and optimism are identified as significant mediators, influencing the relationship between emotion regulation strategies and dimensions of psychological well-being. Significant direct effects of positive reappraisal and planning on self-acceptance and purpose in life were observed, whereas catastrophizing demonstrated negative direct effects on environmental mastery. The indirect effects showed that emotion regulation strategies affect psychological well-being by influencing self-esteem and optimism.

## 4. Discussion

The aim of the study was to examine the relationship between emotion regulation strategies and psychological well-being, as well as to investigate the mediating role of self-esteem and optimism in the period of emerging adulthood. This research is important because it sheds light on how emotion regulation strategies relate to psychological well-being in young adulthood. This, in turn, can provide valuable insights into the factors that lead to positive well-being and the optimal development of an individual’s emotional state.

The first hypothesis was that emotion regulation strategies would be associated with psychological well-being, with adaptive strategies expected to have a positive relationship and maladaptive strategies a negative relationship with dimensions of well-being. Our findings support a significant correlation between adaptive strategies and well-being, so this direction was confirmed. The most significant associations were observed for planning, positive reappraisal, and catastrophizing across most well-being dimensions. The findings suggest that the use of adaptive emotion regulation strategies in response to challenging circumstances is associated with enhanced levels of well-being ([1]; [39]; [43]).

The second hypothesis was a positive connection between emotion regulation, self-esteem, and optimism, which was supported in this study: Adaptive strategies, such as planning and positive reappraisal, demonstrated the strongest correlations with self-esteem and optimism. Conversely, the catastrophizing strategy exhibited the most significant negative correlations with both self-esteem and optimism. In the context of emerging adulthood, a period characterized by instability, identity exploration, and elevated uncertainty, these findings suggest that the use of adaptive strategies could enable individuals to navigate challenges with higher levels of self-confidence and a more positive view of the future ([8]; [68]). Conversely, in the case of maladaptive strategies such as catastrophizing, this could be linked to a diminished self-image and a pessimistic view of the future, which could act as a limiting factor in the individual’s ability to adapt to and respond to the challenges of this life stage ([25]).

The third hypothesis, that self-esteem and optimism would be positively related to psychological well-being, was also confirmed. These findings are consistent with Ryff’s model ([54], [55]), which suggests that a sense of purpose in life is related to self-acceptance and the ability to control one’s environment. In particular, self-esteem can be conceptualized as a resource that strengthens the perception of personal value, thereby facilitating the acceptance of both strengths and weaknesses ([52]). Optimism has been shown to be associated with a greater orientation towards goals or life purposes, especially in a context marked by uncertainty, such as emerging adulthood ([57]).

Finally, the fourth hypothesis was that self-esteem and optimism would mediate the relationship between emotion regulation and well-being. The results indicated that these variables play a crucial role in this relationship, as adaptive strategies contribute to enhanced well-being, whereas maladaptive strategies have the opposite effect. High self-esteem and high levels of optimism enhance the benefits of positive reappraisal, favoring greater self-acceptance, environment mastery, and meaning in life. Therefore, promoting positive self-esteem and optimism can significantly increase the effectiveness of positive reappraisal, thereby enhancing emotion regulation and overall well-being ([68]).

Planning was the emotion regulation strategy with the greatest direct, indirect, and total effects. Furthermore, planning was found to be highly related to optimism and self-esteem. Planning facilitates the development of coping strategies for stressful situations, thereby reducing uncertainty and stress. This, in turn, facilitates a more effective mastery of the environment and an enhanced sense of self-acceptance and life purpose ([30], [31]). The integration of planning, self-esteem, and optimism represents a significant structure in the development of a proactive disposition toward goals. This constructive process not only improves the ability to regulate emotions but also has a beneficial effect on self-esteem and the ability to recognize and accept oneself as one is ([14]; [59]). With respect to positive reappraisal, the adoption of a constructive reinterpretation of a stressful event as a mechanism of emotion regulation has the potential to enhance psychological well-being. This is because it enables individuals to perceive difficulties as opportunities for growth, thereby reinforcing their self-acceptance and sense of control ([33]; [63]).

Self-esteem and optimism act as complete mediators in the relationship between maladaptive emotion regulation strategies such as catastrophizing and the dimensions of psychological well-being. A notable finding of this study is the direct positive effect of catastrophizing on the dimension of life purpose. Catastrophizing involves excessive negative thinking, which may lead to further introspection as people try to understand and manage their emotional responses ([29]). This introspection might result in greater self-awareness; although focused on the negative aspects, it may allow deep reflection on life circumstances, achieving a reappraisal of personal goals and stimulating adaptive change ([60]). When self-esteem and optimism are included as mediating variables, the overall effect of catastrophizing on life purpose changes from positive to negative. While catastrophic thinking may initially prompt introspection regarding the meaning of life, low self-esteem and a negative perceptions about the future can ultimately have a negative impact on life purpose ([58]).

In the context of stressful circumstances that may give rise to catastrophic ideas during emerging adulthood, high self-esteem and optimistic thinking can facilitate a more balanced and positive evaluation of the situation. Conversely, catastrophic beliefs associated with low self-esteem and a pessimistic view of the future could have a significant impact on several aspects of well-being, such as self-acceptance, environmental mastery, and purpose in life. This is because these ideas overestimate one’s limitations, making it difficult to be accepting and compassionate towards oneself, limiting the ability to effectively master the circumstances of daily life and compromising the search for meaning and commitment to important and meaningful goals ([6]; [20]; [51]).

The results presented are significant concerning the challenges associated with emerging adulthood. This period is characterized by a phase of exploration and the consolidation of a coherent personality, which is of great importance to well-being ([49]). The capacity for self-acceptance is of fundamental significance during this phase, as it comprises the identification and acceptance of both personal strengths and weaknesses. Such acceptance allows for the development of a more accurate and resilient self-image, which is a crucial element in emotion development ([49]). For this, self-acceptance at this stage is associated with higher levels of self-esteem, self-confidence, and optimism because these factors are crucial for individuals to be adequately prepared to confront the challenges and uncertainties inherent to this period ([9]; [57]).

Likewise, emerging adulthood is also characterized by a multitude of changes, including the transition to university life, the commencement of one’s career, changes in one’s place of residence, and the setting up of new relationships. Therefore, an adequate level of environmental mastery will allow individuals to effectively manage these transitions and adapt to new situations, which in turn is vital for well-being ([5]; [6]). Moreover, the search for a purpose in life is a central characteristic of emerging adulthood. Individuals explore different career options, relationships, and lifestyles to identify what matters to them and what they consider relevant. It can be seen, therefore, that adaptive emotion regulation is vital for maintaining psychological well-being despite the uncertainties and changes that characterize this stage of life ([57]).

### Strengths, Implications and Limits

The findings of this study have significant practical implications for the development of psychological interventions and educational contexts. The results indicate that psychological interventions may be most effective when they focus on reinforcing self-esteem and fostering optimism. It is recommended that strategies such as positive reappraisal and planning be promoted as a way of enhancing well-being. Self-esteem and optimism may serve to reduce the impact of catastrophizing thoughts on psychological well-being, thereby reducing their negative effects. In the context of higher education, these findings offer a basis for the development of educational resources to foster optimism and self-esteem in university students and to enhance their ability to respond effectively to academic and personal stressors. The implementation of programs of this nature by educational institutions would serve to provide students with the necessary support for their mental health and well-being during this critical developmental stage.

Psychoeducational programs that emphasize emotion regulation and the strengthening of personal resources for well-being should be prioritized within university wellness initiatives, as universities are primary settings serving this population. At the institutional level, these findings also underscore the need for policymakers to prioritize mental health promotion strategies tailored to the needs of emerging adults. Specifically, the results support the development of targeted interventions aimed at fostering socio-emotion competencies, including innovative approaches such as serious games ([66]) and cognitive behavioral techniques adapted for this developmental stage ([27]).

While the study yielded meaningful findings, it is not without limitations. The sample consisted exclusively of university-educated emerging adults, which may restrict the generalizability of the results. However, this focus is contextually relevant, as this life stage is increasingly characterized by extended academic engagement due to the demands of a knowledge- and technology-based economy ([3]). Future research should aim to include non-university emerging adults for comparative analysis and expand the sample to encompass individuals from a wider range of academic disciplines and age brackets. Additionally, efforts should be made to achieve a more balanced gender distribution, as women tend to be overrepresented in psychology-related studies, a trend reflected in our sample.

Another limitation concerns the exclusive use of self-report measures, which are susceptible to biases such as social desirability. Nonetheless, these instruments remain valuable for reaching large samples and capturing subjective psychological experiences. Finally, the cross-sectional nature of the study limits causal inferences. Future research would benefit from longitudinal designs or the inclusion of complementary methods such as behavioral observations, physiological measurements, or external evaluations to complement the findings and provide a more comprehensive perspective on psychological well-being in emerging adulthood.

## 5. Conclusions

Emotion regulation strategies and the development of self-esteem and optimism during emerging adulthood play a crucial role in dimensions of psychological well-being, specifically self-acceptance, environmental mastery, and purpose in life. The capacity to master one’s environment enables individuals to effectively navigate numerous transitions and adapt to new situations, which are significant in the consolidation of their personality, identity and emotional management. Moreover, the optimistic perspective allows emerging adults to maintain a sense of purpose in life and in the face of the uncertainties and changes that characterize this stage of life. Taken together, these dimensions of well-being and emotion regulation strategies contribute significantly to a more purposeful and meaningful life in emerging adulthood. Additionally, this research can enhance the theoretical comprehension of human development during the period of emerging adulthood. By examining the way emotion regulation, optimism, and self-esteem affect well-being, it is possible to develop more comprehensive theories of the processes of emotion during this crucial stage.

## Figures and Tables

**Figure 1 behavsci-15-00929-f001:**
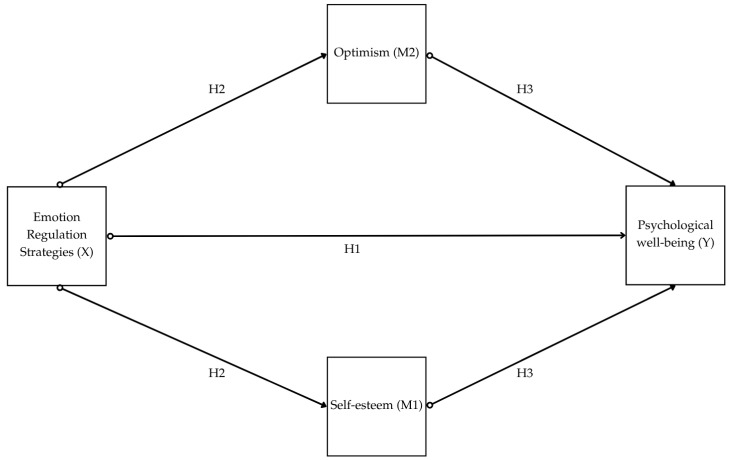
Graphical representation of study’s hypotheses. Note: H1 = Hypothesis 1; H2 = Hypothesis 2; H3 = Hypothesis 3.

**Table 1 behavsci-15-00929-t001:** Descriptive statistics for the variables.

Variable		M	SD	Sk	Ks	Min	Max	*α*	*ω*
Adaptive strategies	Planning	7.67	1.58	−0.48	0.04	2	10	0.65	N/A
	Acceptance	7.98	1.41	−0.52	0.37	2	10	0.73	N/A
	Positive reappraisal	6.39	2.12	−0.02	−0.69	2	10	0.84	N/A
	Putting into perspective	6.94	1.61	0.0	−0.46	2	10	0.51	N/A
	Positive refocusing	5.62	1.93	0.07	−0.53	2	10	0.73	N/A
Maladaptive strategies	Rumination	7.01	1.70	−0.25	−0.34	2	10	0.63	N/A
	Catastrophizing	5.73	1.88	0.18	−0.58	2	10	0.79	N/A
	Self-blame	6.45	1.88	−0.19	−0.32	2	10	0.81	N/A
	Other-blame	4.68	1.65	0.40	0.07	2	10	0.85	N/A
Psychological well-being	Self-acceptance	16.77	4.23	−0.23	−0.52	4	24	0.86	0.86
	Positive relations	23.05	5.16	−0.63	−0.21	7	30	0.81	0.81
	Autonomy	24.10	5.81	−0.31	−0.48	7	36	0.78	0.77
	Environmental mastery	20.45	4.12	−0.08	−0.41	8	30	0.65	0.63
	Purpose in life	20.36	5.24	−0.08	−0.62	6	30	0.85	0.85
	Personal growth	19.55	3.44	−0.57	−0.19	8	24	0.71	0.73
Mediation variables	Self-esteem	31.10	5.72	−0.48	−0.36	11	40	0.90	0.90
	Optimism	35.19	6.76	−0.66	0.13	12	45	0.90	0.90

Note: M = mean; SD = standard deviation; Sk = skewness; Ks = Kurtosis; *α* = Alfa Cronbach; *ω* = Omega; N/A = not applicable. Omega coefficients were not calculated for variables with only three items, due to the fact that Omega requires at least four items in order to produce stable and meaningful estimates.

**Table 2 behavsci-15-00929-t002:** Correlation matrix of the study.

Variable	1	2	3	4	5	6	7	8	9	10	11	12	13	14	15	16
Rumination	—															
Catastrophizing	0.52 ***	—														
Self-blame	0.32 ***	0.27 ***	—													
Other-blame	0.07 *	0.25 ***	−0.10 **	—												
Planning	−0.08 *	−0.27 ***	−0.07 *	0.02	—											
Acceptance	−0.063	−0.24 ***	−0.05	−0.04	0.39 ***	—										
Positive reappraisal	−0.25 ***	−0.45 ***	−0.09 **	−0.13 ***	0.49 ***	0.33 ***	—									
Putting into perspective	−0.12 ***	−0.11 **	0.06	−0.00	0.22 ***	0.30 ***	0.28 ***	—								
Positive refocusing	−0.22 ***	−0.18 ***	−0.14 ***	0.10 **	0.31 ***	0.16 ***	0.32 ***	0.26 ***	—							
Self-acceptance	−0.17 ***	−0.40 ***	−0.22 ***	−0.07 *	0.50 ***	0.32 ***	0.47 ***	0.18 ***	0.27 ***	—						
Positive relations	−0.10 **	−0.24 ***	−0.15 ***	−0.12 ***	0.19 ***	0.08 *	0.19 ***	0.05	0.08 *	0.41 ***	—					
Autonomy	−0.25 ***	−0.35 ***	−0.23 ***	−0.06	0.32 ***	0.17 ***	0.31 ***	0.02	0.14 ***	0.46 ***	0.29 ***	—				
Environmental mastery	−0.23 ***	−0.40 ***	−0.14 ***	−0.15 ***	0.41 ***	0.30 ***	0.40 ***	0.12 ***	0.17 ***	0.72 ***	0.41 ***	0.37 ***	—			
Purpose in life	−0.14 ***	−0.29 ***	−0.14 ***	−0.05	0.47 ***	0.30 ***	0.43 ***	0.17 ***	0.24 ***	0.78 ***	0.28 ***	0.35 ***	0.72 ***	—		
Personal growth	−0.02	−0.22 ***	−0.13 ***	−0.06	0.39 ***	0.26 ***	0.32 ***	0.09 **	0.09 *	0.58 ***	0.30 ***	0.30 ***	0.50 ***	0.54 ***	—	
Self-esteem	−0.23 ***	−0.43 ***	−0.29 ***	−0.11 **	0.43 ***	0.24 ***	0.43 ***	0.10 **	0.22 ***	0.79 ***	0.41 ***	0.51 ***	0.66 ***	0.63 ***	0.53 ***	—
Optimism	−0.27 ***	−0.47 ***	−0.21 ***	−0.12 ***	0.51 ***	0.29 ***	0.53 ***	0.16 ***	0.27 ***	0.75 ***	0.35 ***	0.43 ***	0.67 ***	0.69 ***	0.53 ***	0.78 ***

*Note*: * *p* < 0.05, ** *p* < 0.01, *** *p* < 0.001.

**Table 3 behavsci-15-00929-t003:** Direct, indirect and total effects mediation model.

								Interval	
(X)	(M)	(Y)	Effect Type	Est	Std	Z-Value	*p*	Low	Up
Positive reappraisal	N/A	Self-acceptance	Direct	0.03	0.01	2.08	0.04	0.00	0.05
Positive reappraisal	N/A	Environmental mastery	Direct	0.01	0.02	0.43	0.66	−0.02	0.04
Positive reappraisal	N/A	Purpose in life	Direct	0.03	0.02	2.12	0.03	0.00	0.06
Planning	N/A	Self-acceptance	Direct	0.07	0.02	4.70	<0.00	0.04	0.10
Planning	N/A	Environmental mastery	Direct	0.04	0.02	2.19	0.03	0.00	0.08
Planning	N/A	Purpose in life	Direct	0.08	0.02	4.10	<0.00	0.04	0.12
Catastrophizing	N/A	Self-acceptance	Direct	0.00	0.01	−0.10	0.91	−0.03	0.02
Catastrophizing	N/A	Environmental mastery	Direct	−0.04	0.02	−2.23	0.02	−0.07	−0.03
Catastrophizing	N/A	Purpose in life	Direct	0.05	0.02	3.28	0.00	0.02	0.08
Positive reappraisal	Self-esteem	Self-acceptance	Indirect	0.04	0.01	4.57	<0.00	0.02	0.06
Positive reappraisal	Optimism	Self-acceptance	Indirect	0.03	0.01	5.06	<0.00	0.02	0.05
Positive reappraisal	Self-esteem	Environmental mastery	Indirect	0.03	0.01	4.16	<0.00	0.01	0.04
Positive reappraisal	Optimism	Environmental mastery	Indirect	0.04	0.01	5.22	<0.00	0.03	0.06
Positive reappraisal	Self-esteem	Purpose in life	Indirect	0.02	0.01	3.96	<0.00	0.01	0.03
Positive reappraisal	Optimism	Purpose in life	Indirect	0.05	0.01	6.11	<0.00	0.03	0.07
Planning	Self-esteem	Self-acceptance	Indirect	0.09	0.01	6.65	<0.00	0.06	0.11
Planning	Optimism	Self-acceptance	Indirect	0.05	0.01	5.64	<0.00	0.04	0.07
Planning	Self-esteem	Environmental mastery	Indirect	0.06	0.01	5.18	<0.00	0.04	0.08
Planning	Optimism	Environmental mastery	Indirect	0.07	0.01	5.70	<0.00	0.05	0.10
Planning	Self-esteem	Purpose in life	Indirect	0.04	0.01	4.52	<0.00	0.03	0.06
Planning	Optimism	Purpose in life	Indirect	0.09	0.01	7.37	<0.00	0.07	0.12
Catastrophizing	Self-esteem	Self-acceptance	Indirect	−0.08	0.01	−7.61	<0.00	−0.10	−0.06
Catastrophizing	Optimism	Self-acceptance	Indirect	−0.04	0.01	−5.35	<0.00	−0.05	−0.03
Catastrophizing	Self-esteem	Environmental mastery	Indirect	−0.05	0.01	−5.73	<0.00	−0.06	−0.03
Catastrophizing	Optimism	Environmental mastery	Indirect	−0.05	0.01	−5.54	<0.00	−0.07	−0.03
Catastrophizing	Self-esteem	Purpose in life	Indirect	−0.04	0.01	−4.96	<0.00	−0.05	−0.02
Catastrophizing	Optimism	Purpose in life	Indirect	−0.07	0.01	−6.76	<0.00	−0.09	−0.05
Positive reappraisal	N/A	Self-acceptance	Total	0.10	0.02	5.90	<0.00	0.07	0.13
Positive reappraisal	N/A	Environmental mastery	Total	0.07	0.02	4.20	<0.00	0.04	0.11
Positive reappraisal	N/A	Purpose in life	Total	0.10	0.02	5.96	<0.00	0.07	0.14
Planning	N/A	Self-acceptance	Total	0.21	0.02	10.16	<0.00	0.17	0.25
Planning	N/A	Environmental mastery	Total	0.16	0.02	7.50	<0.00	0.12	0.21
Planning	N/A	Purpose in life	Total	0.21	0.02	9.63	<0.00	0.17	0.26
Catastrophizing	N/A	Self-acceptance	Total	−0.11	0.02	−6.72	<0.00	−0.15	−0.08
Catastrophizing	N/A	Environmental mastery	Total	−0.13	0.02	−7.36	<0.00	−0.17	−0.10
Catastrophizing	N/A	Purpose in life	Total	−0.05	0.02	−2.78	0.00	−0.09	−0.08

Note: X = Emotion regulation strategies; M = Mediator; Y = Psychological well-being dimension; Est = Estimate; Std = Standard deviation.

## Data Availability

The raw data supporting the conclusions of this article will be made available by the authors on request.

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
