# Peer review of "Emotion Regulation Strategies and Psychological Well-Being in Emerging Adulthood: Mediating Role of Optimism and Self-Esteem in a University Student Sample"

_behavsci, 2025, doi:10.3390/bs15070929_

Round 1

Reviewer 1 Report

Comments and Suggestions for Authors

Thank you for the opportunity to review an article that addresses a very important topic. I have read the study with interest and congratulate the authors on their quality work. Below are a few specific observations on particular sections that came to mind, which I offer to the authors as an opportunity to consider what else might strengthen the quality of their study

INTRODUCTION

The introduction effectively provides a solid theoretical foundation for the study, highlighting the importance of emerging adulthood, emotion regulation, self-esteem, and optimism. The explanation of adaptive and maladaptive emotion regulation strategies is clear and well-supported by literature.

However, while the theoretical grounding is strong, the introduction could benefit from a clearer structure. Specifically, the definition of emerging adulthood, which appears on lines 62 onward, should be introduced earlier in the text. This would allow the authors to present the studied areas in direct connection with the concept of emerging adulthood, providing a more logical and coherent introduction.

Moreover, while the theoretical grounding is strong, the introduction could benefit from a more explicit rationale for why this study is necessary, particularly in terms of its contribution to the existing body of knowledge. Thus, after reading the introduction, the reader already has the impression that "that individuals with higher self-esteem and a positive optimist on the future are more likely to engage in adaptive emotion regulation strategies, which, in turn, enhances their well-being" is perfectly logical and self-evident. Similarly proposed hypotheses. It would be useful to point out in more depth the necessity and importance of the research and how it enriches current knowledge.

MATERIALS AND METHODS

Again, the section is well described and provides a detailed and methodologically sound description. I appreciate well-explained sampling process, with clear criteria for inclusion and exclusion, ensuring that the final sample is appropriately refined.

I have some minor questions:

I just have a few minor questions: how was the questionnaire distributed and how were participants recruited? Did they receive any remuneration for participating in the study? How long did it take the participants to complete?

Interestingly, no socio-demographic data is given. Were these collected? And why are they not presented, e.g. was there no difference in results between genders (73% female?!) or different ages/years of study?

This is also related to the following observation: the sampling method is described as convenience sampling from various universities. I think this may limit the generalizability of the results. The high proportion of undergraduate students may also introduce a sampling bias. A more diverse sample could enhance the study's external validity. Or is it common in Spain for all emerging adults to graduate from university? In this context, it might be appropriate to modify the full title of the article to state that it is only about university students.

Additionally, while the description of statistical methods is detailed, a more explicit explanation of why these specific instruments were chosen for measuring the main variables would strengthen the methodological justification.

RESULTS

Congrats authors on this part, it is well structured and data are clearly presented from descriptive statistics and correlations to the mediation model and its direct, indirect, and total effects.  Only in a few places could the authors consider a more accessible description of the results presented in the table

 Although the mediation model fit indices are presented, a more explicit justification for accepting a CFI value below .90 as adequate could be added, especially since some guidelines recommend a higher threshold (CFI > .90).

The correlation analysis is extensive, but it would benefit from a more focused summary highlighting the most relevant relationships. While the authors mention correlations above .3, a brief discussion of the strongest and weakest correlations among the variables would enhance the understanding, or at least drawing attention to these specific results.

DISCUSSION AND CONCLUSION

This section effectively summarizes the key findings, including the confirmation of the four hypotheses. The linkage with prior research enhances the theoretical grounding of the article when the authors connect these findings with existing literature.  

In this part I would appreciate a deeper interpretaion of unexpected results, which are an interesting part of the study. The finding that catastrophizing has a positive direct effect on the purpose in life is mentioned, but the interpretation is minimal. For example, is it possible that experiencing negative emotions can sometimes lead an individual to reflect on the meaning of life or to seek meaning in adversity? In this context, it might be appropriate to consider the spiritual orientation of individuals or their overall value orientation.

Similarly, contextualising the results could strengthen the discussion. Although the authors discuss the relevance of the findings to adolescence, they could have better emphasized how these results apply specifically to this developmental stage, rather than to well-being in general, as highlighting how young adults uniquely face emotional challenges should be central to the article.

The authors acknowledge the limitation of using a convenience sample of university students, which may restrict generalizability which I appreciate given my comment above. They may also state other limitations, such as the reliance on self-report measures, which can introduce social desirability bias. And also they should mention the study's cross-sectional design limits conclusions about causal relationships.

The summarising and practical recommendations are commendable. The emphasis on practical applications aligns well with the study's focus on emerging adulthood.

Overall, I find the article useful and very nicely presented. I thank the authors for it and I thank them for their work in this important area. I hope that the above comments will contribute to making their study of interest to the general public.

Author Response

Response to Reviewer 1

Comment 1: Define emerging adulthood earlier in the introduction

Response 1: The definition of emerging adulthood is now introduced earlier in the text, in the new subsection 1.1 Emerging adulthood: A new developmental stage (lines 38–62), to provide a clearer foundation before presenting the theoretical framework.

Comment 2: Clarify the rationale and contribution of the study

Response 2: We added a paragraph at the end of the introduction (lines 146–152) to explain why this study is relevant and how it contributes to a better understanding of emotional mechanisms in emerging adulthood.

Comment 3: Explain how the questionnaire was distributed, how participants were recruited, incentives, and duration

Response 3: This information was added in lines 192–195. It explains that professors shared the questionnaire with students, sometimes in class. Participants completed it online in approximately 20 minutes and were entered into a raffle for Amazon gift cards.

Comment 4: Sociodemographic data and sample characteristics

Response 4: These aspects was added in lines (185 -191). Also, in the Strengths and limitations section, we acknowledge the gender imbalance and discuss how the predominance of psychology students may affect generalizability.

Comment 5: Sampling method and generalizability; clarify if all emerging adults in Spain are university students; adjust the title.

Response 5: We address this limitation in lines 479–481. The manuscript title has been revised to reflect the specific population: “ in a University Student Sample”.

Comment 6: Justify the choice of instruments

Response 6: Lines 201–203 now include a rationale for selecting the instruments, based on their psychometric properties, prior use in similar populations, and fit with the study's objectives.

Comments 7: Justify CFI below .90

Response 7: we include justification for accepting a CFI of .90 as adequate. (lines 304 – 311)

Comments 8. Highlight strongest and weakest correlations

Response 8: A focused summary of key correlations has been added in lines 284–291, identifying the most relevant relationships to improve interpretation. In addition, the tables have been improved to make them easier to read.

Comment 9: Expand interpretation of unexpected results (direct positive effect between catastrophism and life purpose)

Response 9: We add the discussion in lines 413–423. How reappraisal of personal goals could stimulate adaptive changes.

Comment 10: Contextualize findings specifically in emerging adulthood

Response 10: Lines 434–455 emphasize the developmental context and explain how emotional regulation and personal resources operate uniquely during this stage.

Comment 11: Mention additional limitations (self-report bias, cross-sectional design)

Response11: These are discussed in lines 477–493 as part of the Strengths and limitations section.

Comment 12: Positive evaluation of the practical focus

Response 12: We expanded the practical applications in lines 469–476, including examples like psychoeducation, cognitive-behavioral training, serious games (Velert, 2025), and institutional strategies for mental health promotion (Gangemi, 2019).

Reviewer 2 Report

Comments and Suggestions for Authors

This manuscript presents an interesting and timely contribution to the understanding of emotional regulation, cognitive distortions, and personality traits in young adulthood. The study is methodologically sound, clearly written, and addresses relevant psychological constructs with practical implications. I particularly appreciated the strength of the methodology section. However, I believe that the manuscript would benefit from improved organization and the addition of some conceptual clarifications to enhance clarity, fluency, and critical depth. 

  1. Introduction Enhancement

    The introduction is clear and well written. However, it would benefit from the inclusion of background literature and data specifically focused on young adulthood, such as previous studies on well-being, optimism, or psychological development in this age group. Currently, some of this content appears only in the discussion section, but it would be more appropriate and effective to include it earlier, in the introduction, to frame the relevance of the study population.

    To improve clarity and structure, I suggest organizing the introduction into two subsections:

    • One brief specifically dedicated to describing the target population (young adults), including relevant literature and contextual data;

    • Another focused on emotional strategies, personality traits, and other constructs, setting the theoretical foundation for the constructs investigated.

    This structure would help readers follow the rationale behind the study and its relevance more effectively. 

  2. Clarify Study Aims and Hypotheses
    I also recommend creating a dedicated subsection to clearly state the aims and hypotheses of the study. This will help guide the reader and improve the structural flow of the manuscript.

  3. Test for Normality
    Please include the test for normal distribution (e.g., Kolmogorov–Smirnov or Shapiro–Wilk test) to confirm that the use of Pearson’s correlation was appropriate. This would strengthen the methodological rigor of the statistical analysis.

  4. Clarify Constructs from the Start
    From the beginning of the paper, it is important to distinguish clearly between:

    • Emotion regulation strategies (e.g., coping)

    • Cognitive distortions (e.g., catastrophizing)

    • Personality traits (e.g., optimism)

    While the title and early sections emphasize emotion regulation and personality, the analysis also explores cognitive distortions and maladaptive strategies (e.g., catastrophizing or rumination). I suggest either refining the introduction and title to reflect this broader scope or clearly explaining the conceptual overlap early in the text.

  5. Expand and Structure the Discussion
    The discussion could benefit from more structure. I suggest adding a subsection on the implications of the findings. This could include concrete examples of how institutions or policymakers might support the well-being of young adults (e.g., through psychoeducation, peer support programs, access to therapy, anti-stigma policies, etc.). Currently, some of this content appears only in the conclusion section, but it would be more appropriate and effective to include it in the discussions. You can also add to the discussion examples of specific CBT trainings, such as Gangemi et al. (2019). Reducing Probability Overestimation of Threatening Events: A Study on the Efficacy of Cognitive Techniques, Clinical Neuropsychiatry, 16(3):149–155.

  6. Highlight the Role of Cognitive Distortions
    I also recommend emphasizing the potential consequences of cognitive distortions and maladaptive strategies, as these patterns may increase the risk for more serious psychopathological outcomes. A relevant reference could be: Gangemi et al. (2021). Emotional Reasoning and Psychopathology, Brain Sciences, 11, 471.

  7. Add a Section on Strengths and Limitations
    Consider creating a distinct section to discuss the strengths and limitations of the study. This will help provide a balanced, critical view and is often expected in empirical research papers.

  8. Conclusions should not report implications of the study, but only a summary of the main findings. 

Author Response

Response to Reviewer 2

Comments 1: Introduction enhancement: include more literature on emerging adulthood and organizing into two subsections.

Response 1: We added more background literature on the developmental stage of emerging adulthood (e.g., Arnett, 2018; Baggio et al., 2017; Brito & Soares, 2023; Mahmoud et al., 2012).

The introduction has been restructured into distinct subsections:

1.1 Emerging adulthood: A new developmental stage (lines 39–62)

1.2 Psychological well-being in emerging adulthood (lines 64–88)

1.3 Emotion regulation strategies and mental health (lines 90–112)

1.4 Personality traits: self-esteem and optimism (lines 114–144)

Comments 2: Clarify aims and hypotheses

Response 2: A new subsection has been added: 1.5 Objective and hypothesis of the study (lines 146–169), which clearly states the study's aim and four hypotheses.

Comments 3: Include a normality test

Response 3: We conducted the Kolmogorov–Smirnov test and reviewed histograms and Q-Q plots. The results are reported in lines 259–262, confirming the use of Pearson’s correlation.

Comments 4: Clarify constructs from the start

Response 4: The constructs of emotion regulation strategies, and dispositional traits (self-esteem, optimism) are clearly distinguished in the new structure of the introduction and their conceptual connections are explained.

Comments 5: Expand and Structure the Discussion

Response 5: A new subsection was added in the discussion (lines 457–493) focusing on practical implications. Also, we include specific suggestions such as psychoeducational programs, peer networks, and mental health promotion in university contexts. Likewise we also refer to CBT techniques such as those described in Gangemi et al. (2019).

Comments 6: Highlight the role of cognitive distortions and their consequences

Response 6: The potential effects of maladaptive strategies are discussed in lines 370–378. We included a reference to Gangemi et al. (2021).

Comments 7: Add a Section on Strengths and Limitations

Response 7: A new section titled 4.1 Strengths and limitations was added, (lines 447- 493) discussing the sample size, use of validated instruments, and limitations related to self-report measures, gender imbalance, and cross-sectional design.

Comment 8: Conclusions should not report implications of the study, but only a summary of the main findings.

Response 8: The Conclusions section (lines 494–508) now focuses only on the main findings of the study. Content related to practical applications was moved to the discussion.

Round 2

Reviewer 1 Report

Comments and Suggestions for Authors

Thank you for addressing my comments

Reviewer 2 Report

Comments and Suggestions for Authors

The paper has shown clear improvement since the previous version. The revisions have addressed several important points, and the manuscript is more coherent and focused. However, a few minor issues still need to be addressed before final acceptance.

  1. There are still some minor editorial/ typographical errors throughout the manuscript. Careful proofreading is recommended.

  2. The statement in lines 417–418 lacks clarity in terms of its scientific status. It is not clear whether the authors are presenting a hypothesis, an empirical conclusion, or a general assumption. If this is part of the study’s main thesis or interpretation, the phrasing should be adjusted to reflect an appropriate level of certainty (e.g., using terms like “may”, “might”, or “seem to”). If it is an established claim or supported by previous research, a relevant citation should be added to substantiate it.

  3. The paragraph addressing the implications or future directions of the study is located within the “Limits and Strengths” section (Section 4.1). For clarity and structure, it would be preferable to either:
    a) Create a new subsection within the Discussion section specifically titled “Implications and Future Directions,” or
    b) Revise the title of Section 4.1 to more accurately reflect its expanded scope (e.g., “Limits, Strengths, and Implications”).
    This change would help distinguish between methodological considerations and broader theoretical or practical implications, enhancing the overall organization of the discussion.

Author Response

Comments 1. Editorial/typographical errors

Response 1: We have revised the manuscript to correct grammatical, typographical, and stylistic issues. In this process, we paid particular attention to punctuation, lexical consistency and appropriate academic phrasing. All changes are highlighted in the document.

Comments 2:  The statement in lines 417–418 lacks clarity in terms of its scientific status

Response 2: The statement has been revised to reflect an appropriate level of certainty using modal verbs such as may and might, as suggested. Additionally, we have included a new citation (Seager, 2009) to support this interpretation. These changes are highlighted in the document.

Comments 3: Section title: “Limits and Strengths”

Response 3: In accordance with the reviewer’s suggestion, we have revised the title of section 4.1 to: “Strengths, Implications, and Limits”. These changes are highlighted in the document.

Round 3

Reviewer 2 Report

Comments and Suggestions for Authors

The paper is fine and ready for publication in my opinion